# miR-125b-5p, miR-155-3p, and miR-214-5p and Target *E2F2* Gene in Oral Squamous Cell Carcinoma

**DOI:** 10.3390/ijms24076320

**Published:** 2023-03-28

**Authors:** Karolina Gołąbek, Dorota Hudy, Agata Świętek, Jadwiga Gaździcka, Natalia Dąbrowska, Katarzyna Miśkiewicz-Orczyk, Natalia Zięba, Maciej Misiołek, Joanna Katarzyna Strzelczyk

**Affiliations:** 1Department of Medical and Molecular Biology, Faculty of Medical Sciences in Zabrze, Medical University of Silesia in Katowice, 19 Jordana St., 41-808 Zabrze, Poland; 2Silesia LabMed Research and Implementation Center, Medical University of Silesia in Katowice, 19 Jordana St., 41-808 Zabrze, Poland; 3Strathclyde Institute of Pharmacy and Biomedical Sciences, University of Strathclyde, 161 Cathedral Street, Glasgow G4 0RE, UK; 4Department of Otorhinolaryngology and Oncological Laryngology, Faculty of Medical Sciences in Zabrze, Medical University of Silesia in Katowice, 10 C Skłodowska St., 41-800 Zabrze, Poland

**Keywords:** oral squamous cell carcinoma, OSCC, E2F2, miR-125b-5p, miR-155-3p, miR-214-5p, tumour, margin

## Abstract

It is known that E2F2 (E2F transcription factor 2) plays an important role as controller in the cell cycle. This study aimed to analyse the expression of the *E2F2* gene and E2F2 protein and demonstrate *E2F2* target microRNAs (miRNAs) candidates (miR-125b-5p, miR-155-3p, and miR-214-5p) in oral squamous cell carcinoma tumour and margin samples. The study group consisted 50 patients. The *E2F2* gene and miRNAs expression levels were assessed by qPCR, while the E2F2 protein was assessed by ELISA. When analysing the effect of miRNAs expression on *E2F2* gene expression and E2F2 protein level, we observed no statistically significant correlations. miR-125b-5p was downregulated, while miR-155-3p, and miR-214-5p were upregulated in tumour samples compared to margin. We observed a difference between the miR-125b-5p expression level in smokers and non-smokers in margin samples. Furthermore, HPV-positive individuals had a significantly higher miR-125b-5p and miR-214-5p expression level compared to HPV-negative patients in tumour samples. The study result showed that the *E2F2* gene is not the target for analysed miRNAs in OSCC. Moreover, miR-155-3p and miR-125b-5p could play roles in the pathogenesis of OSCC. A differential expression of the analysed miRNAs was observed in response to tobacco smoke and HPV status.

## 1. Introduction

Oral squamous cell carcinoma (OSCC) is considered one of the most common cancers in the head and neck squamous cell carcinoma (HNSCC) region. More than 300,000 new cases were reported worldwide in 2020 according to GLOBOCAN 2020 analysis [1]. It is also worth adding that this type of cancer is characterized by a 5 year survival rate ranging only from 45 to 50% [2,3]. It is believed that 80% of squamous cell carcinoma cases are due to alcohol consumption and smoking as risk factors [4,5]. However, the genetic background of this cancer type seems still insufficiently understood. Therefore, it seems reasonable to search for new biomarkers for an improved diagnosis and prognosis of the disease.

The E2F family of transcription factors includes eight proteins (E2F1–E2F8). These factors play an important role in cell proliferation and cell-cycle progression, regulate autophagy, mitochondrial functions, and the DNA damage response [6,7]. Pocket proteins, such as hypophosphorylated retinoblastoma protein Rb, and related proteins p107 and p130 can critically control E2F activity [6]. Changes in the *E2F* gene expression were analysed in several types of cancer—breast cancer, gastric cancer, liver cancer, and non-small-cell lung cancer [6,8,9,10,11]. The higher *E2F2* expression levels were significantly associated with poor prognosis in patients with non-small-cell lung cancer [6,8].

MicroRNAs (miRNAs) are a class of small, non-coding RNAs with about 18–22 nucleotides. The action mechanism of miRNA is based on binding to the 3′-UTR of mRNAs and silencing them [12,13]. These types of RNA are involved in the post-transcriptional regulation of approximately 60% of human genes [14]. miRNAs play an important role in modulating processes such as embryogenesis, cellular development, cell proliferation, cell metabolism, and homeostasis [13]. The importance of miRNAs in cancer biology was first demonstrated by Calin et al. [15] in a study to determine frequent deletions and downregulation of miR15 and miR16 in chronic lymphocytic leukaemia [15]. Currently, many studies on miRNAs focus on how these molecules affect gene expression in carcinogenesis. miRNAs are tissue-specific and may be seen as oncogenic (OncomiRs) when involved in the repression of tumour suppressor gene expression. In addition, miRNAs as tumour suppressors (oncosuppressor miRs) can also participate in the downregulation of oncogenes [16,17]. Some studies examine the role of miRNA in various cancers, such as pancreatic [18], colon [19], prostate [20,21], renal [22], bladder [23], breast [24], cervical [25], ovarian [26], lung [27], and HNSCC [28]. To date, analyses of the role of miRNAs in HNSCC pathogenesis are focused generally on studies of miRNAs, such as miR-375, miR-1234, miR-103, miR-638, miR-200b-3p, miR-191-5p, miR-24-3p, miR-572, miR-483-5p, miR-20a, miR-22, miR-29a, miR-29b, mir-let-7c, miR-17, miR-374b-5p, miR-425-5p, miR-122, miR-134, miR-184, miR 191, miR-412, miR-512, miR-8392, miR-21, miR-31, miR-155, miR-196a, miR-196b, miR-9, miR-29c, miR-223, miR-187, Let-7a, miR- 27, miR-34, miR-92, miR-124, miR-125a, miR-136, miR-139, miR-145, miR-146a, miR-200, miR-195, and miR-205 [29,30,31,32,33].

This study aimed to analyse the expression of the *E2F2* gene and E2F2 protein and demonstrate *E2F2* target miRNA candidates (miR-125b-5p, miR-155-3p, and miR-214-5p) in oral squamous cell carcinoma tumour and margin samples.

## 2. Results

### 2.1. Patient Sociodemographic and Clinical Characteristics

In the patient cohort group, the median age was 62.5 years (range: 27–87 years). There were 38 (76%) men and 12 (24%) women; 27 (54%) reported alcohol consumption; 28 (56%) patients who were smokers; and 17 (34%) who were both smokers and alcohol users. HPV 16 DNA was found in 13 of 50 samples (26%). The 3 year survival rate for all patients was approximately 24%. Clinical parameters of the OSCC group are shown in Table 1. 

### 2.2. E2F2 Gene Expression and E2F2 Protein Expression in Tumour Samples Compared to Margin Samples

We found no statistically significant differences in the *E2F2* expression gene level and E2F2 protein level in tumour samples compared to margin samples. The median *E2F2* relative gene expression in the tumour was 0.263 (0.145–0.578) and 0.47 (0.051–0.784) in the margin. While the median E2F2 protein level in the tumour was 0.129 ng/µg (0.051–0.238) and 0.082 ng/µg (0.047–0.282) in the margin (Figure 1). We also showed no effect of gene expression on protein expression in the examined tissues. 

### 2.3. miR-125b-5p, miR-155-3p, and miR-214-5p Expression in Tumour Samples Compared to Margin Samples

Analysis of the relative expression levels of the tested miRNAs demonstrated statistically significant levels for only one (miR-155-3p) of the tested miRNAs when comparing tumour versus margin samples. The tests showed that miR-125b-5p (*p*-value = 0.235) was downregulated, while miR-155-3p (*p*-value = 0.013), and miR-214-5p (*p*-value = 0.368) were upregulated in tumour samples compared to the margin samples (Figure 2).

### 2.4. Correlation of E2F2 Gene Expression and E2F2 Protein Concentration with miRNAs Expression 

When analysing the effect of miRNAs expression on *E2F2* gene expression and E2PF2 protein level, we observed no statistically significant difference in the tumour samples. We showed a similar observation in margin samples. Figure 3 presents the correlation between the expressions of *E2F2* gene, E2F2 protein, and the tested miRNAs in tumour and margin samples. 

### 2.5. Correlations between E2F2 Gene Expression and Sociodemographic and Clinicopathological Features

No association was found between the *E2F2* gene expression levels, age, gender, smoking, alcohol consumption, HPV status and clinical parameters (N and G) in tumour and margin samples, except T parameter. The patients with T1 had a significantly higher gene expression level of *E2F2* than patients with T2 (0.581 vs. 0.237; *p*-value = 0.019) and T3 (0.581 vs. 0.145; *p*-value = 0.004) in tumour samples. Furthermore, we noticed that high *E2F2* gene expression levels in the margin could contribute to a lower 3 year survival rate (*p*-value = 0.026).

### 2.6. Correlations between E2F2 Protein Expression and Sociodemographic and Clinicopathological Features

No association was found between the E2F2 protein expression levels, age, gender, smoking, alcohol consumption, and clinical parameters (T and G) in tumour and margin samples, except N parameter. The patients with N0 had a significantly lower protein level of E2F2 than patients with N3 (0.125 vs. 0.581; *p*-value = 0.022) in tumour samples. At the same time, protein expression was lower in HPV-positive individuals compared to HPV-negative patients (0.035 vs. 0.164; *p*-value = 0.002) in tumour samples (Figure 4).

### 2.7. Correlation of miRNAs Expression Level with Sociodemographic and Clinicopathological Variables

We also assessed the correlation of the expression levels of selected miRNAs with demographics and clinical characteristics of the patients. When analysing the effect of smoking on miRNAs expression, we observed a statistically significant difference between the miR-125b-5p expression level in smokers and non-smokers (0.568 vs. 2.194; *p*-value = 0.009) in margin samples. In addition, the patients with G2 had a significantly lower miR-125b-5p expression level than patients with G3 (0.507 vs. 0.992, *p*-value = 0.008) in tumour samples. Furthermore, HPV-positive individuals had significantly higher miR-125b-5p and miR-214-5p expression levels in tumour samples compared to HPV-negative patients (2.629 vs. 0.627; *p*-value = 0.035 and 7.942 vs 1.684; *p*-value = 0.039, respectively). Figure 5, Figure 6 and Figure 7 present the expressions of the tested miRNAs in HPV-positive compared to HPV-negative patients.

## 3. Discussion

It is known that E2F2 plays an important role as a controller in the cell cycle [6]. Our study demonstrated no statistically significant differences in the *E2F2* expression gene level and the E2F2 protein level in tumour samples compared to margin samples. Still, we found a higher expression of *E2F2* gene in the margin samples, while E2F2 protein level was higher in the tumour specimens. Furthermore, we noticed that the tumour progression could be associated with a decrease in gene expression. High *E2F2* gene expression levels in the margin could contribute to a lower 3 year survival rate. Moreover, the patients with N0 had a significantly lower protein expression level of E2F2 than patients with N3 in tumour samples. At the same time, protein concentration was lower in HPV-positive individuals compared to HPV-negative patients in tumour samples. Perhaps these results are related to the ambiguous role of E2F2 as a transcription factor in cell cycle regulation. This protein can act as both a promoter of cell division and a suppressor. Furthermore, changes in gene expression profile in normal samples could be explained by Slaughter’s model of “field cancerization”. According to this idea, in the samples of the margin without a trace of the tumour, cells genetically altered by the action of carcinogens could be present [34,35]. It is also noteworthy that E7 is a well-known HPV oncogene that plays a critical role in cellular growth control pathways. E7 targets the retinoblastoma protein Rb of tumour suppressor for proteasome-mediated degradation, and this protein can critically control the E2F family of transcription factors activity [6,36].

For many years, there has been interest in research with regard to the use of miRNAs as potential diagnostic and prognostic biomarkers in many types of cancers. Based on several studies, it can be assumed that miRNAs (for example microRNA-let-7a, miR-125b, miR-146b-3p, miR-638, miR-31, miR-218, miR-454-3p and miRNA-936) could play an important role in targeting *E2F2* expression. The analyses concerned with types of cancer, such as breast cancer, ovarian cancer, colon cancer, gastric cancer, prostate cancer, hepatocellular carcinoma, non-small cell lung cancer cell, glioma, and laryngeal cancer, suggested that miRNAs, by targeting *E2F2*, could be involved in cancer progression, overall survival of patients, and response to radiochemotherapy [37,38,39,40,41,42,43,44,45,46]. Moreover, circ_RPPH1/miR-146b-3p/*E2F2* axis could promote the progression of breast cancer and circCUL2/miR-214-5p/*E2F2* axis-suppressed retinoblastoma cells [46,47]. When analysing the effect of miRNAs expression on *E2F2* gene expression and E2F2 protein concentration, we observed no statistically significant differences in the OSCC tumour samples. However, we showed a similar observation in margin samples.

The miR-125 family is highly conserved and composed of a few homologs (e.g., miR-125a-3p, miR-125a-5p, miR-125b-1 and miR-125b-2) [48]. Some studies showed miR-125b could be downregulated in many types of tumours. Decreased miRNA expression was found in carcinomas of bladder [49], breast [50,51], liver [52,53], ovary [54,55], as well as Ewing’s sarcoma [56]. Upregulation of miR-125b is also known to reverse the effects of drug resistance in ovarian cancer cells and nasopharyngeal carcinoma cells [57,58]. In this work, we demonstrated that miR-125b-5p was downregulated in tumour samples but not at statistically significant levels. However, it is noteworthy that the patients with G2 had a significantly lower miR-125b-5p expression level than patients with G3 in tumour samples. This result confirms previous analyses. The study by Shiiba [59] and others has shown a downregulated expression of miR-125b in OSCC-derived cell lines and OSCC samples. Furthermore, forced expression of miR-125b has been observed to enhance radiosensitivity in OSCC cells. Moreover, miR-125b expression is correlated with OSCC tumour staging and survival [59]. Our tests showed a statistically significant difference between the miR-125b-5p expression level in smokers and non-smokers in margin samples. The Doukas et al. [60] analysis demonstrated that tobacco could play a role in miRNA expression modification in HNSCC. The tobacco-specific nitrosamine—NNK (4-(methylnitrosamino)-1-(3-pyridyl)-1-butanone) has been identified as an important inducing factor in the upregulation of miR-21 and miR-155 and in the downregulation of miR-422 [60]. Perhaps miR-125b-5p expression is also regulated by NNK. Furthermore, HPV-positive individuals had a significantly higher miR-125b-5p expression level in tumour samples compared to HPV-negative patients. Our observation thus confirms the result of another study. The analysis of miRNA expression in cells transfected with HPV 11, 16, and 45 showed that this virus could be a modifier of miRNAs expression. It is noteworthy that the most upregulated miRNA is miR-125a-5p in cancers such as head and neck and gastric cancer [61].

The miR-155 host gene (MIR155HG) produces two different miRNA strands: miR155-5p and miR-155-3p [62]. In our study, we showed a statistically significant increased expression of miR-155-3p in tumour samples compared to margin samples. Several studies reported higher miR-155-5p expression in OSCC tumour samples [63,64,65]. Furthermore, miR-155-5p overexpression was associated with ESCC (oesophageal squamous cell carcinoma) tumour aggressiveness [66] and lymph node metastases and relapse OSCC [64,65]. It is worth adding that a meta-analysis based on eight studies including 709 HNSCC patients (279 presenting OSCC) showed that the OSSC subgroup had a worsening survival for patients with a high expression of miR-155-5p in tumour samples [33]. 

In our study, the last of the miRNAs analysed was miR-214-5p. miR-214-5p is known as a tumour-suppressive miRNA that plays a role in the pathogenesis of cancers such as osteosarcoma [67], hepatocellular carcinoma [68], and pancreatic cancer [69]. It is also noteworthy that miR-214-5p was upregulated in the tongue squamous cell carcinoma cisplatin-resistant subline [70]. Yao and others [71] demonstrated that miR-214-5p expression had a negative correlation with *E2F2* expression in pancreatic cancer tissue samples [71]. Our study showed no association between miR-214-5p level and *E2F2* expression, but HPV-positive individuals had a significantly higher miR-214-5p expression compared to HPV-negative patients in tumour samples. However, the role of miR-214-5p in HPV replication has not been reported. Interestingly, Patil et al.’s [72] study indicated that miR-214-5p regulates viral replications of hepatitis E virus [72].

In summary, one of the main limitations of this study is the small sample size. In addition, our results should be confirmed in studies using OSCC cell lines.

## 4. Materials and Methods

### 4.1. Patient and Samples

In total, 50 paired tumour and matching margin specimens were collected from patients with OSCC. The tissues were obtained following surgical resection at the Department of Otorhinolaryngology and Oncological Laryngology, Faculty of Medical Sciences in Zabrze, Medical University of Silesia in Katowice. Tumour staging was based on the American Joint Committee on Cancer (AJCC, version 2007) [73,74] and the WHO Classification of Head and Neck Tumours [75]. The normal tissues (margins) were checked and classified as cancer-free by pathologists. The main inclusion criteria were: written informed consent to participate in the study, age over 18 years, no metabolic diseases (e.g., diabetes, hypertension) or no chronic inflammatory diseases, a diagnosis of primary tumours, and no history of preoperative radio- or chemotherapy. All laboratory analyses were performed at the Department of Medical and Molecular Biology, Faculty of Medical Sciences in Zabrze, Medical University of Silesia in Katowice. The samples were transported to the Department on ice. The study was approved by the Bioethics Committee of the Medical University of Silesia (approval no. KNW/022/KB1/49/16 and no. KNW/002/KB1/49/II/16/17). 

### 4.2. RNA and miRNAs Extraction and Quantification

The methodology for the extraction was presented in a previous study [76]. All tissue samples were homogenized with FastPrep^®^-24 homogenizer (MP Biomedicals, Solon, CA, USA) with ceramic beads Lysing Matrix D (MP Biomedicals, Solon, CA, USA). The RNA and miRNAs were extracted using the RNA isolation kit (catalog number RIK 001, BioVendor, Brno, Czech Republic) to the standard instruction. The concentration and purity of the isolated RNA was determined using spectrophotometry in NanoPhotometer Pearl UV/Vis Spectrophotometer (Implen, Munich, Germany) [76].

### 4.3. Selection of Candidate MicroRNAs to E2F2 Target

The target miRNAs of the *E2F2* gene were predicted by miRCode (version 11) [77], miRDB (version 6.0) [78], and TargetScan (version 7.2) [79] online databases.

### 4.4. Complementary DNA (cDNA) Synthesis

The methodology for the cDNA synthesis was presented in a previous study [76]. Total RNA (5 ng) was reverse-transcribed into cDNA using a high capacity cDNA reverse transcription kit with RNase inhibitor (Applied Biosystems, Foster City, CA, USA) according to manufacturer’s protocol. The compound mix was prepared in a 20 μL volume containing: 2 μL of 10× Buffer RT; 0.8 μL of 25× dNTP mix (100 mM); 2 μL of 10× RT random primers; 1 μL of MultiScribe^®^ reverse transcriptase; 1 μL of RNase inhibitor; 3.2 μL of nuclease-free H_2_O; and 10 μL of previously isolated RNA. The reaction was processed in Mastercycler personal (Eppendorf, Hamburg, Germany) with the following thermal profile: 25 °C for 10 min, 37 °C for 120 min, 85 °C for 5 min, and 4 °C–∞ [76].

Furthermore, the obtained RNA (5 ng) was reverse-transcribed using TaqMan^®^ Advanced miRNA cDNA synthesis kit (Applied Biosystems, Foster City, CA, USA) according to manufacturers’ protocol. The whole procedure consisted four reactions with different profiles, which are presented in Table 2. The reactions were prepared in Mastercycler personal (Eppendorf, Hamburg, Germany).

### 4.5. E2F2 Gene and miRNAs Expression Analysis

The methodology for the *E2F2* gene expression analysis was presented in a previous study [35]. The relative gene expression (RQ) analysis was performed by Real-Time PCR (qPCR) using TaqMan^®^ gene expression assays, QuantStudio 5 Real-Time PCR System, and Analysis Software v1.5.1 (Applied Biosystems, Foster City, CA, USA). The kit was supplied with primers and fluorescently marked molecular probes. The glyceraldehyde-3-phosphate dehydrogenase gene (*GAPDH*) was used as an endogenous control. Five surgical margin samples were used as a calibrator. The comparative threshold cycle (Ct) method 2^−∆∆Ct^ was used to determine the RQ. The qPCR was performed in a volume of 20 µL using 1 µL of cDNA, 10 µL of TaqMan^®^ fast advanced master mix (Applied Biosystems, Foster City, CA, USA), 1 µL of TaqMan^®^ gene expression assays (Assay ID: Hs00914334_m1 for *E2F2*, and Assay ID: Hs03929097_g1 for *GAPDH*), and 8 µL of nuclease-free H_2_O (EURx, Gdańsk, Poland). Thermal cycle for all analysed genes was: 95 °C for 20 s, followed by 40 cycles of 95 °C for 1 s, and 60 °C for 20 s [76].

The relative expression (RQ) of miR-125b-5p, miR-155-3p, and miR-214-5p were assessed based on the guidelines by the TaqMan^®^ advanced miRNA assays (Applied Biosystems, Foster City, CA, USA). The kit was supplied with primers and fluorescently marked molecular probes. All reactions were performed in QuantStudio 5 Real-Time PCR System and Analysis Software v1.5.1 (Applied Biosystems, Foster City, CA, USA). The housekeeping miR-361-5p was used for normalizing the expression. Five surgical margin samples were used as a calibrator. RQ was calculated using 2^−∆∆Ct^ after normalization with the reference miRNA. The qPCR was performed in a volume of 20 µL using 5 µL of cDNA (1:10 dilution), 10 µL of TaqMan^®^ fast advanced master mix (Applied Biosystems, Foster City, CA, USA) and 1 µL of TaqMan^®^ advanced miRNA assay (assay ID: 477885_mir for miR-125b-5p, assay ID: 477926_mir for miR-155-3p, assay ID: 478768_mir for miR-214-5p, and assay ID: 478056_mir for miR-361-5p). Thermal cycle for all analysed miRNAs was: 95 °C for 20 s, followed by 40 cycles of 95 °C for 1 s, and 60 °C for 20 s. Table 3 presents the sequence of the tested miRNAs.

### 4.6. E2F2 and Total Protein Concentration Determinations

The tumour and margin tissue samples were homogenized with a homogenizer Bio-Gen PRO200 (PRO Scientific Inc., Oxford, CT, USA) at a speed 10,000 RPM (5 times 1 min at 2 min intervals) in nine volumes of cold PBS (EURx, Gdańsk, Poland). Then, the suspensions were sonicated with the ultrasonic processor UP100H (Hilscher, Teltow, Germany). 

Enzyme-linked immunosorbent assay (ELISA) technique was used to evaluate E2F2 protein concentrations in tissue homogenates. Enzyme-linked immunosorbent assay kit (Assay ID: SEJ182Hu, Cloud-Clone Corp., Houston, TX, USA) was used for the tests according to the manufacturer’s instructions. Absorbance was recorded at 450 nm wavelength and calibrated according to the standard curve in Synergy H1 microplate reader (BioTek, Winooski, VT, USA) and results were calculated with Gen5 2.06 software (BioTek, Winooski, VT, USA). The sensitivity of this assay was 0.059 ng/mL. Precision-measured as the coefficient of variation was <10% (intra-assay) and <12% (inter-assay). 

The quantified total protein was performed using AccuOrange™ protein quantitation kit (Biotium, Fremont, CA, USA) according to the standard instructions. The samples were assayed in a 100-fold dilution. The detection range of the assay was 0.1–15 µg/mL protein. The fluorescence was measured with excitation/emission at 480/598 nm (SYNERGY H1 microplate reader; BIOTEK, Winooski, VT, USA). The E2F2 protein concentrations for each sample were normalized to the total amount of protein in the tissue lysates. Values were expressed in ng/µg protein.

### 4.7. HPV 16 Detection

DNA was extracted from tissue samples using a gene matrix tissue DNA purification kit (EURx, Gdansk, Poland) according to the manufacturer’s instructions. The concentration and purity of the isolated DNA was prepared using spectrophotometry in NanoPhotometer Pearl UV/Vis Spectrophotometer (Implen, Munich, Germany). HPV was detected using AmpliSens^®^ HPV 16/18-FRT PCR kit (InterLabService, Moscow, Russia) according to manufacturers’ protocol. The PCR was performed in a volume of 25 µL using 7 µL of PCR-mix-1-FEP/FRT HPV; 8 µL mixture of PCR-buffer-FRT, TaqF polymerase, and 10 µL of DNA. The amplification program was as follows: 95 °C for 20 s, followed by 45 cycles of 95 °C for 20 s, and 60 °C for 1 min. All PCR reactions were performed using in QuantStudio 5 Real-Time PCR System (Applied Biosystems, Foster City, CA, USA).

### 4.8. Statistical Analyses

The Shapiro–Wilk test evaluated the distribution of variables. The median with inter-quartile range (25–75%) was used to describe miRNAs expression, *E2F2* gene expression, and E2F2 protein concentration. The Mann–Whitney U test was used to compare the sociodemographic and clinical characteristics, miRNAs, *E2F2* gene, and E2F2 protein level. Correlations between miRNAs, *E2F2* gene, and E2F2 protein concentration were calculated using Spearman’s rank correlation analysis. The level of statistical significance was set at 0.05. The statistical software STATISTICA version 13 (TIBCO Software Inc., Palo Alto, CA, USA) was used to perform all the analyses.

## 5. Conclusions

The study result showed that the *E2F2* gene is not the target for analysing miRNAs in OSCC. Moreover, miR-155-3p and miR-125b-5p could play roles in the pathogenesis of OSCC. A differential expression of the analysed miRNAs in response to tobacco smoke and HPV status was observed.

## Figures and Tables

**Figure 1 ijms-24-06320-f001:**
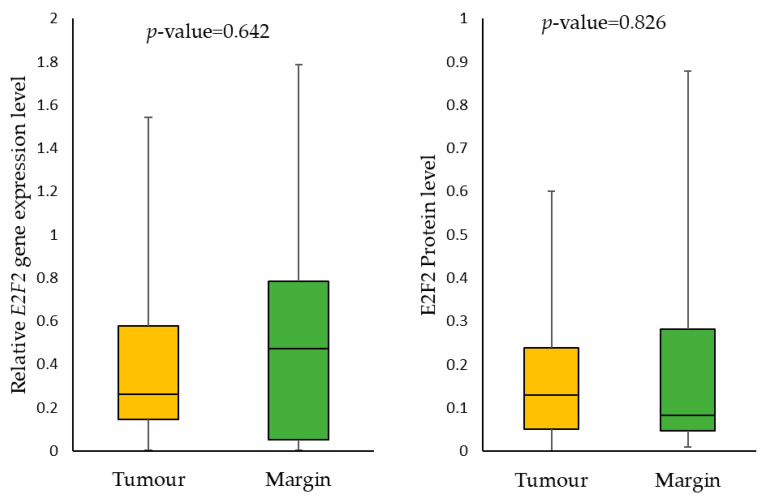
The *E2F2* mRNA expression and E2F2 protein level in tumour and margin samples.

**Figure 2 ijms-24-06320-f002:**
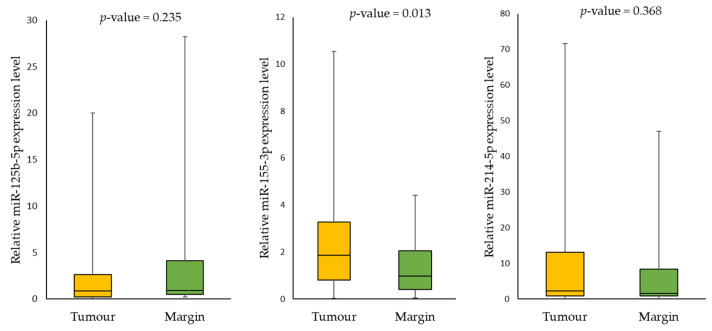
The relative expression levels of tested miRNAs in tumour samples compared to margin samples.

**Figure 3 ijms-24-06320-f003:**
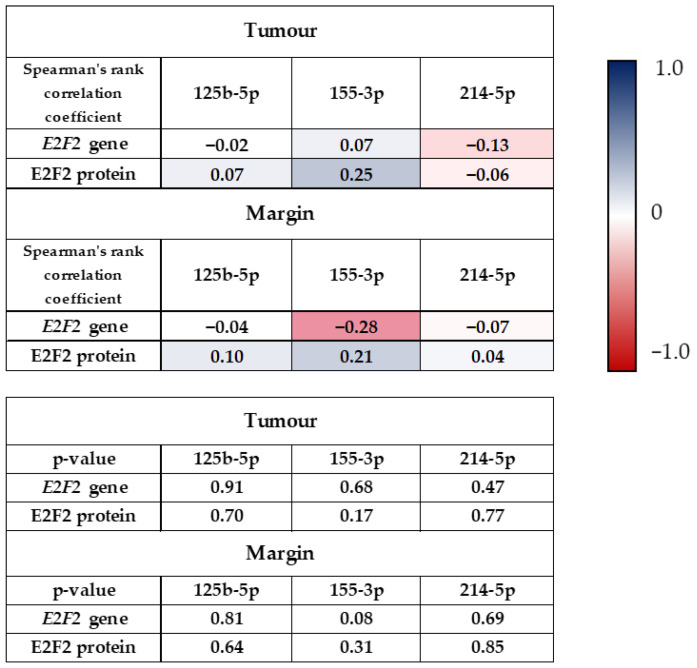
Correlation between the expressions of *E2F2* gene, E2F2 protein, and the tested miRNAs in tumour and margin samples.

**Figure 4 ijms-24-06320-f004:**
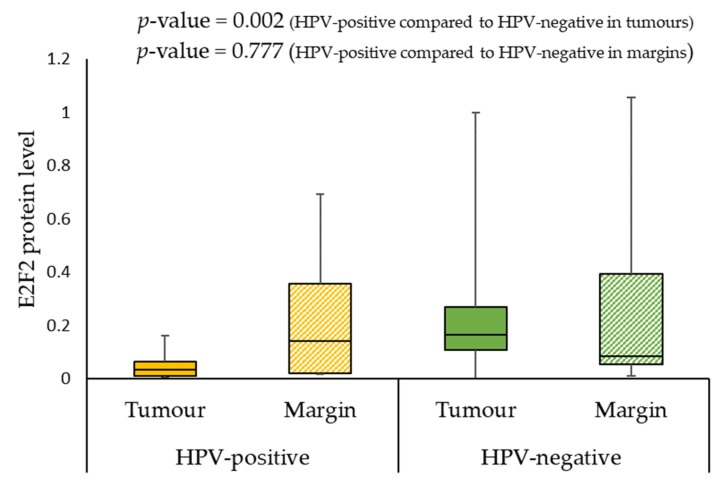
E2F2 protein level in HPV-positive compared to HPV-negative patients.

**Figure 5 ijms-24-06320-f005:**
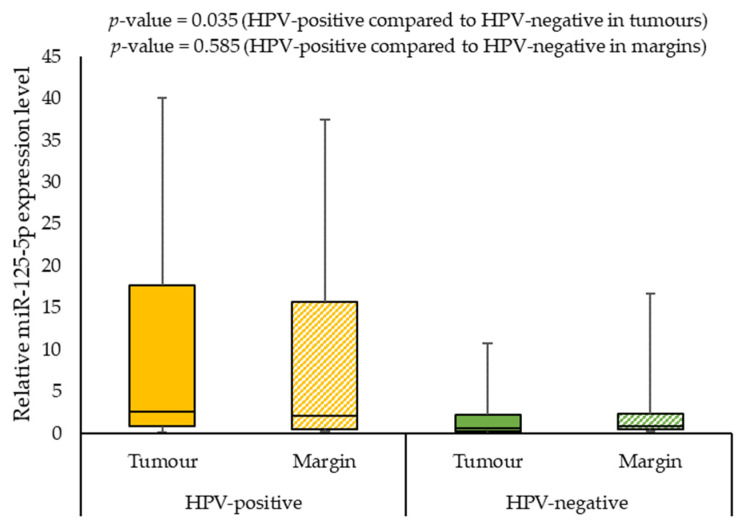
The relative expression level of miR-125b-5p in HPV-positive compared to HPV-negative patients.

**Figure 6 ijms-24-06320-f006:**
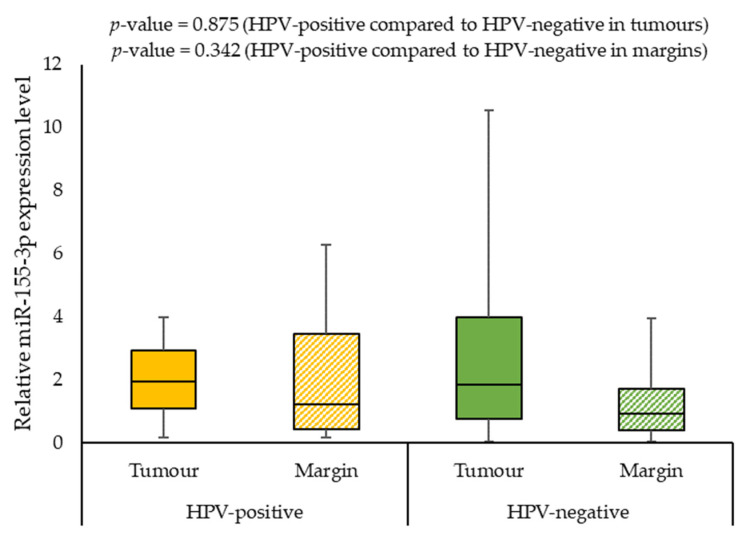
The relative expression level of miR-155-3p in HPV-positive compared to HPV-negative patients.

**Figure 7 ijms-24-06320-f007:**
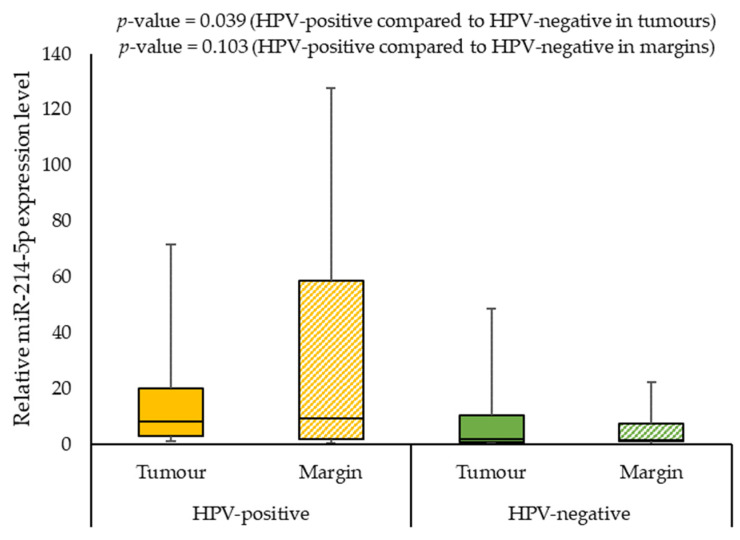
The relative expression level of miR-214-5p in HPV-positive compared to HPV-negative patients.

**Table 1 ijms-24-06320-t001:** Clinical parameters of the OSCC group.

Clinical Parameters	Patients, *n* (%)
Histological grading	
G1 (Well differentiated)	9 (18)
G2 (Moderately differentiated)	23 (46)
G3 (Poorly differentiated)	18 (36)
T classification	
T1	10 (20)
T2	23 (46)
T3	16 (32)
T4	1 (2)
Nodal status	
N0	24 (48)
N1	2 (4)
N2	20 (40)
N3	4 (8)
Patient status at 3 years	
Alive	12 (24)
Dead	38 (76)

**Table 2 ijms-24-06320-t002:** cDNA synthesis procedure using TaqMan^®^ Advanced miRNA cDNA synthesis kit (Applied Biosystems, Foster City, CA, USA) according to manufacturer’s protocol.

Perform the poly(A) tailing reaction	**Reaction Composition**			
	**Component**	**Volume**		
	10× Poly(A) Buffer	0.5 μL		
	ATP	0.5 μL		
	Poly(A) enzyme	0.3 μL		
	RNA sample	2 μL		
	RNase-free water	1.7 μL		
	**Thermal Profile**			
	**Step**	**Temperature**	**Time**	
	Polyadenylation	37 °C	45 min	
	Stop reaction	65 °C	10 min	
	Hold	4 °C	Hold	
Perform the adaptor ligation reaction	**Reaction Composition**			
	**Component**	**Volume**		
	5× DNA Ligase Buffer	3 μL		
	50% PEG 8000	4.5 μL		
	25× Ligation Adaptor	0.6 μL		
	RNA Ligase	1.5 μL		
	Poly(A) tailing reaction product	5 μL		
	RNase-free water	0.4 μL		
	**Thermal Profile**			
	**Step**	**Temperature**	**Time**	
	Ligation	16 °C	60 min	
	Hold	4 °C	Hold	
Perform the reverse transcription (RT) reaction	**Reaction Composition**			
	**Component**	**Volume**		
	5× RT Buffer	6 μL		
	dNTP Mix (25 mM each)	1.2 μL		
	20× Universal RT primer	1.5 μL		
	10× RT enzyme mix	3 μL		
	Adaptorligation reaction product	15 μL		
	RNase-free water	3.3 μL		
	**Thermal Profile**			
	**Step**	**Temperature**	**Time**	
	Reverse transcription	42 °C	15 min	
	Stop reaction	85 °C	5 min	
	Hold	4 °C	Hold	
Perform the miR-Amp reaction	**Reaction Composition**			
	**Component**	**Volume**		
	2× miR-Amp master mix	25 μL		
	20× miR-Amp primer mix	2.5 μL		
	RT reaction product	5 μL		
	RNase-free water	17.5 μL		
	**Thermal Profile**			
	**Step**	**Temperature**	**Time**	**Cycles**
	Enzyme activation	95 °C	5 min	1
	Denature	95 °C	3 s	14
	Anneal/extend	60 °C	30 s	
	Stop reaction	99 °C	10 min	1
	Hold	4 °C	Hold	1

**Table 3 ijms-24-06320-t003:** Sequences of analysed miRNAs.

miRNA	Mature miRNA Sequence
miR-125b-5p	UCCCUGAGACCCUAACUUGUGA
miR-155-3p	CUCCUACAUAUUAGCAUUAACA
miR-214-5p	UGCCUGUCUACACUUGCUGUGC
miR-361-5p (Housekeeping control)	UUAUCAGAAUCUCCAGGGGUAC

## Data Availability

The data used to support the findings of this study are available from the corresponding author upon request.

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
