# Peer review of "miR-125b-5p, miR-155-3p, and miR-214-5p and Target E2F2 Gene in Oral Squamous Cell Carcinoma"

_ijms, 2023, doi:10.3390/ijms24076320_

Round 1
Reviewer 1 Report
In this article, the authors aimed to analyze the expression of the E2F2 gene and protein and identify miRNA candidates in oral squamous cell carcinoma (OSCC) tumor and margin samples. Their results showed no significant correlations between miRNA expression and E2F2 gene or protein expression. However, miR-125b-5p was downregulated, while miR-155-3p and miR-214-5p were upregulated in tumor samples compared to margin. Moreover, miR-155-3p and miR-125b-5p could play roles in the pathogenesis of OSCC. The study also observed a differential expression of the analyzed miRNAs in response to tobacco smoke and HPV status. However, there are some remaining questions to be answered:
1, For relative E2F2 gene expression and protein level, could the authors described how they calculated the values? Any normalization? It seems like neither tumour nor Margin group has been normalized to 1 in Figure 1.
2, What is the unit for y-axis for Relative gene expression level if the data were not normalized?
3, The authors used 3 software (miRCode, miRDB, and TargetScan) to predict miRNA for E2F2? Why the authors chose to select the predicted miRNA for evaluation? Any experimental evidence showed the direction relationship between these miRNAs and E2F2?Are these 3 miRNAs the only miRNA targets?
4, Figure 4, what is the p-values? The tumor and margin in HPV-positive and negative group? Or tumours/Margins samples between HPV-positive and negative group?
5, Line 147-150. They didn’t seem to be part of the manuscript.
6, Any previous study evaluated the predictive role of E2F2 in OSCC or HNSCC? Did they examined the mRNA or protein level? Since E2F2 plays a role in cell proliferation and cell cycle? Should the authors examined the proliferation marker (For instance, Ki67) or cell cycle markers in tumour and margin samples?
Reviewer 2 Report
Karolina Gołąbek's manuscript aimed to analyze the expression of the E2F2 gene and E2F2 protein 18 and to demonstrate target miRNA candidates in oral squamous cell carcinomas. We observed a difference between the expression level of miR-125b-5p 25 in smokers and non-smokers. Furthermore, HPV-positive individuals had an unequally higher expression level of miR-125b-5p and miR-214-5p than HPV-negative patients.
I congratulate the authors for the excellent work done. The manuscript is in fact well-structured in all its parts. The conclusions also reflect what was previously expressed by the authors.
Just to underline the importance of mRNAs in oral squamous cell carcinoma, I recommend adding a few more references in the introduction.
I suggest two manuscript (DOI: 10.3390/jpm13020275; doi: 10.3390/biology1105651) to be placed on line 74.
Reviewer 3 Report
miRNAs are now often studied as markers in various cancers, including HNC.The Authors analysed the expression of the miR-125b-5p, miR-155-3p, and miR-214-5p and E2F2 gene and E2F2 protein in OSCC and margin samples.
Authors pointed the E2F2 gene is not the target for analysing miRNAs in OSCC.
Minor comments
1. E2F2 protein level in HPV-positive and HPV-negative patients is presented in Fig.4. In Results authors wrote that miR-125b-5p expression was significantly higher in HPV-positive individuals than in HPV-negative. And was there a difference in the levels of the other two miRNAs in the HPV+ group? It could also be represented graphically.
2. Finally, it might be worth adding whether this study has any clinical, diagnostic or prognostic significance or not.
